# Arbuscular Mycorrhizal Fungal Communities in the Soils of Desert Habitats

**DOI:** 10.3390/microorganisms9020229

**Published:** 2021-01-22

**Authors:** Martti Vasar, John Davison, Siim-Kaarel Sepp, Maarja Öpik, Mari Moora, Kadri Koorem, Yiming Meng, Jane Oja, Asem A. Akhmetzhanova, Saleh Al-Quraishy, Vladimir G. Onipchenko, Juan J. Cantero, Sydney I. Glassman, Wael N. Hozzein, Martin Zobel

**Affiliations:** 1Institute of Ecology and Earth Sciences, University of Tartu, 51005 Tartu, Estonia; martti.vasar@ut.ee (M.V.); john.davison@ut.ee (J.D.); siimsepp@ut.ee (S.-K.S.); maarja.opik@ut.ee (M.Ö.); mari.moora@ut.ee (M.M.); kadri.koorem@ut.ee (K.K.); jane.oja@ut.ee (J.O.); 2Department of Ecology and Plant Geography, Lomonosov Moscow State University, 119234 Moscow, Russia; assemok@mail.ru (A.A.A.); vonipchenko@mail.ru (V.G.O.); 3Zoology Department, College of Science, King Saud University, Riyadh 11451, Saudi Arabia; squraishy@KSU.EDU.SA (S.A.-Q.); whozzein@ksu.edu.sa (W.N.H.); martin.zobel@ut.ee (M.Z.); 4Instituto Multidisciplinario de Biología Vegetal, Universidad Nacional de Córdoba, CONICET, Córdoba 5000, Argentina; juanjocantero@gmail.com; 5Departamento de Biología Agrícola, Facultad de Agronomía y Veterinaria, Universidad Nacional de Río Cuarto, Río Cuarto 5804, Argentina; 6Department of Microbiology and Plant Pathology, University of California, Riverside, CA 92521, USA; sydney.glassman@ucr.edu; 7Botany and Microbiology Department, Faculty of Science, Beni-Suef University, Beni-Suef 62511, Egypt; 8Department of Botany, University of Tartu, Tartu 51006, Estonia

**Keywords:** assembly rules, dryland, fungal community, fungal diversity, mycorrhiza

## Abstract

Deserts cover a significant proportion of the Earth’s surface and continue to expand as a consequence of climate change. Mutualistic arbuscular mycorrhizal (AM) fungi are functionally important plant root symbionts, and may be particularly important in drought stressed systems such as deserts. Here we provide a first molecular characterization of the AM fungi occurring in several desert ecosystems worldwide. We sequenced AM fungal DNA from soil samples collected from deserts in six different regions of the globe using the primer pair WANDA-AML2 with Illumina MiSeq. We recorded altogether 50 AM fungal phylotypes. *Glomeraceae* was the most common family, while *Claroideoglomeraceae, Diversisporaceae* and *Acaulosporaceae* were represented with lower frequency and abundance. The most diverse site, with 35 virtual taxa (VT), was in the Israeli Negev desert. Sites representing harsh conditions yielded relatively few reads and low richness estimates, for example, a Saudi Arabian desert site where only three *Diversispora* VT were recorded. The AM fungal taxa recorded in the desert soils are mostly geographically and ecologically widespread. However, in four sites out of six, communities comprised more desert-affiliated taxa (according to the Maarj*AM* database) than expected at random. AM fungal VT present in samples were phylogenetically clustered compared with the global taxon pool, suggesting that nonrandom assembly processes, notably habitat filtering, may have shaped desert fungal assemblages.

## 1. Introduction

Drylands already cover about 41% of the Earth’s surface [1], and climate change—primarily by increasing aridity and temperature—and intensive land use exacerbate the risk of land degradation and desertification in the near future [2,3]. Indeed, drylands are predicted to cover half of terrestrial Earth’s surface by 2100 even under a moderate emissions scenario [4].

Plant communities in desert areas are unique in many respects. Desert plants have biochemical, physiological and morphological adaptations allowing them to tolerate dry and warm conditions [5,6]. Some of these adaptations may be linked to mutualistic interactions. In particular, mycorrhizal fungi help plants tolerate stressful conditions [7]. Arbuscular mycorrhizal (AM) fungi (phylum Glomeromycota; [8]) are an ancient group of root symbionts that associate with more than 80% of plants in terrestrial ecosystems, gaining plant-assimilated carbon while supplying their hosts with nutrients (mainly phosphorus) and improving their tolerance to abiotic stress and pathogens [9]. AM fungi are found on all continents and many species-level phylogroups (phylogenetically defined groupings of taxa described by DNA sequences) exhibit wide distributions [10].

Arbuscular mycorrhizal fungi are generally thought to help desert plants tolerate stress [11,12], although aridity can reduce overall AM fungal abundance [13]. AM fungi can promote plant drought resistance by producing hyphae with access to small soil pores, greatly increasing the capacity for belowground water uptake [14]. However, AM fungal benefits to hosts may decline under extremely low water availability, as dry soil conditions can inhibit the flow of phosphorus from AM fungi to plants [15]. Indeed, in some extremely arid and nutrient poor areas, non-mycorrhizal plants are more abundant [16,17]. However, a meta-analysis showed that under experimental drought conditions, grasses colonized by AM fungi generally tend to grow larger than those without mycorrhizal symbionts [18]. There is also evidence that water-limited plants grown in arid soils allocate relatively more biomass to AM fungi [19].

While some studies have elucidated how AM fungi can themselves tolerate drought and in turn help their plant partners tolerate drought stress [20,21], the diversity, distribution and ecology of AM fungal communities occurring in arid conditions remains poorly understood. Experimental reduction of rainfall for nearly four months did not result in obvious changes to AM fungal community composition and diversity [22]. While it is true that small organisms tend to tolerate extremely dry conditions better than large ones [23], AM fungal spores are very large among fungi, being up to 1000 times larger than spores of the Ascomycota [24]. Thus, while it is likely that there are fewer constraints on AM fungi than on the plant communities occurring in arid conditions, they might still be more constrained than other microbial taxa such as bacteria and the vast majority of fungi, which have smaller spores. Indeed, spore-based studies have shown relatively low AM fungal diversity in desert regions, with several fungal taxa unique to deserts and other generally widespread taxa missing [25]. AM fungal diversity in Oman was found to be as low as two taxa in a sand dune habitat, but as high as fifteen taxa in more benign desert locations, such as date palm plantations [25], and as high as forty-four taxa in a desert in northwestern China [26]. A DNA-based analysis revealed from nine to eighteen AM fungal phylotypes in the roots of plants growing in a 30 by 30 m plot in a desert ecosystem in Australia [10], another DNA based analyses found ten AM fungal phylotypes in *Vachellia pachyceras* roots from a Kuwaiti desert [27].

Harsh growth conditions for either plants or fungi may both directly influence the composition of desert AM fungal communities and represent a barrier to fungal dispersal. Indeed, ectomycorrhizal fungi show strong biogeographic structure at continental scales [28,29], while dispersal limitation may be weaker for AM fungi at global [30] and regional scales [31]. Data on wind dispersal of AM fungi are mixed [32], while animals have been shown to act as dispersal vectors for AM fungi [33,34,35]. In sparsely vegetated deserts with low densities of both host plants and vector animals, dispersal may not be effective and thus, AM fungal occurrence in desert landscapes may be patchy.

The aim of this study was to gain a preliminary overview of the diversity and species composition of AM fungal communities in different desert ecosystems worldwide based on analysis of DNA extracted from soil. We wanted to sample the full pool of AM fungal taxa, and soil sampling is the straightforward way to do this. We compared soil AM fungal communities in six different desert sites spanning six countries and four continents. We hypothesized that the diversity of AM fungi in desert ecosystems is low and varies between deserts based on edaphic conditions. We also hypothesized that desert communities comprise distinctive taxa that are uncommon in other biomes.

## 2. Methods

### 2.1. Data Collection

Soil for molecular identification of AM fungi was sampled in desert areas in northwestern Argentina, central Australia, southern Israel, southeastern Kazakhstan, central Saudi Arabia and the southwestern United States of America (Figure 1, Table 1 and Table 2). Vegetation varied from very sparsely (<5%) vegetated desert to woody shrubland with vegetation cover over 60%. In each desert location, we identified the site that was least disturbed by human activities. In each sampling site, about 20 g of (0–5 cm) topsoil was collected from twenty randomly located points within approximately a 50 by 50 m sampling area. For further analysis the samples were pooled per site, resulting in an approximately 300 g soil sample for each site. The soil samples were dried within 24 h using silica gel at room temperature and then carefully homogenized. A 2 g subsample of soil was collected from the whole sample for further molecular analysis; the remainder was stored for geochemical analysis.

The climatic variables mean annual temperature and mean annual precipitation were taken from the CHELSA database [38]. Information on the photosynthetic pathway is based on sources [39,40,41,42,43,44]. Information about the previously recorded distribution of individual AM fungal taxa was taken from the Maarj*AM* database, which classifies the central part of published Glomeromycota small-subunit (SSU) rRNA gene sequences into phylogenetically delimited sequence clusters—virtual taxa (VT) [45,46]. Virtual taxa are phylogenetically-defined Operational Taxonomic Units (OTUs) with the approximate resolution of morphologically-defined AM fungal species. For taxa identified from desert samples, this information was used to assess in which biomes they had been previously recorded.

### 2.2. Soil Analyses

We determined soil pH, total N and organic C, as well as content of plant available P, K, Mg and Ca in soils. All soil samples were air dried after sampling at room temperature and thereafter sieved using a sieve with 2 mm openings (Retsch, Haan, Germany). Soil pH was measured in 1M KCl solution following ISO 10390:2005 using the Mettler Toledo pH meter Seven Easy with electrode Mettler Toledo InLab Expert Pro. The content of total N in soil was determined using the Kjeldahl method [47] with the digestion block DK-20 and distillation unit UDK-126 produced by Velp Scientifica Srl (Usmate, Italy). For the determination of organic carbon content in the soil Tjurin’s method was used, in which oxidation was provided by boiling soil samples in sulfuric acid + K_2_Cr_2_O_7_ solution [48]. For determination of soil plant available P, the Mehlich III extraction method was used [49]. The content of elements in the Mehlich III extract was determined using microwave plasma atomic emission spectrometer MP-4200 (Agilent Technologies, Santa Clara, CA, USA). Chemical analyses were performed at the Institute of Agricultural and Environmental Sciences, Estonian University of Life Sciences, Tartu, Estonia.

### 2.3. Molecular Methods and Bioinformatics

DNA was extracted from 5 g of dried soil using a PowerMax^®^ Soil DNA Isolation Kit (MoBio Laboratories, Carlsbad, CA, USA), with the following modifications described by [50]: (1) bead solution tubes were shaken at 60 ˚C for 10 min at 100 rpm in the shaking incubator, and (2) the samples were dried for 10 min at room temperature under a fume hood before adding the final elution buffer. We chose to use the SSU marker region since it is widely used in AM fungal community surveys [46] and a rich database of SSU-based phylogroup diversity is already available [45]. It also exhibits good amplification of most AM fungal families and suitable barcode properties compared with other available marker regions, such as the Internal Transcribed Spacer (ITS) regions [51,52,53]. AM fungal sequences were amplified from soil DNA extracts using AM fungal specific primers for the SSU ribosomal RNA gene: WANDA [54] and AML2 [55]. A first PCR was conducted with amplicon specific primers linked to Illumina Nextera XT sequencing adapters (Illumina forward primer adaptor: 5′-TCGTCGGCAGCGTCAGATGTGTATAAGAGACAG-3′; Illumina reverse primer adaptor: 5′-GTCTCGTGGGCTCGGAGATGTGTATAAGAGACAG-3′). The PCR mixture contained 5 µL of 5XHOT FirePol Blend Master Mix (Solis Biodyne, Tartu, Estonia); 0.5 µL of each 20 µM primer; 1 µL of template DNA and nuclease-free water to reach a total reaction volume of 25 µL. The PCR was performed under the following cycling conditions: 95 °C for 3 min, 40 cycles of 95 °C for 30 s, 55 °C for 30 s, 72 °C for 30 s followed by 72 °C for 5 min. A second PCR was performed with Nextera XT index-adapters. The reaction mix contained 15 µL KAPA HiFi Hotstart PCR mix; 5 µL of Nextera XT index 1 Primer (N7xxx); 5 µL of Nextera XT index 2 Primer (E5xxx); 5 µL of DNA (10 ng µL^−1^) and MQ water to reach a total reaction volume of 30 µL. PCR was performed under following cycling conditions: 95 °C for 3 min, seven cycles of 95 °C for 30 s, 55 °C for 30 s, 72 °C for 30 s followed by 72 °C for 5 min. After the second PCR, samples were purified with Agencourt AMPure XP beads and pooled. The resulting mix was sequenced on the Illumina MiSeq platform, using a 2 × 300 bp paired-read sequencing approach, at Asper Biogene (Tartu, Estonia).

Demultiplexed paired-end reads were analyzed following the bioinformatics steps provided by Vasar et al. [56]. Primer sequences were matched allowing 1 mismatch for both pairs and primers were removed from the paired-end sequences. After removal of barcode and primer sequences, only pairs where both reads had an average quality score of ≥30 were retained. Quality filtered paired-end reads were combined into pairs using FLASh (v1.2.10, [57]) with default parameters (10–300 bp overlap with at least 75% identity). Orphan reads (paired-end reads, where one pair had low average quality or primer mismatch) and unpaired reads (paired-end reads that did not meet the conditions to be combined) were removed from the analyses. Vsearch (v2.14.1, [58]) was used to remove putative chimeric reads using the default parameters and the Maarj*AM* database [45] as a reference set. Retained reads were subjected to a BLAST search (v 2.10.0, [59]) against VT in the Maarj*AM* database using 97% identity and 95% alignment length thresholds. Cultured taxa were defined as those VT that contain sequences of known morphospecies identity according to the Maarj*AM* database [45].

Raw reads from this targeted locus study have been deposited in the NCBI SRA (BioProject PRJNA659159), and representative sequences of each VT have been deposited in the NCBI GenBank under the accession number KELL00000000.

### 2.4. Statistical Analysis

Sampling intensity was assessed using rarefaction curves (the *rarefy* function from the R package [60] vegan [61]). In order to account for differences in sampling intensity between sites, we used the Shannon and Simpson index-based effective numbers of species and extrapolation to an asymptote implemented in the iNEXT software [62]. The asymptotic diversity equates to expected local diversity at full sample coverage. This approach makes it possible to maximize use of the information in the original data, which would be lost with rarefying approaches where many observations are removed.

To address phylogenetic community composition, we used a phylogenetic tree constructed using neighbor-joining with *nj* function from the ape package in R [63] containing the type sequences of all VT in the Maarj*AM* database [64]. We additionally constructed a phylogenetic tree (Appendix A) of the aligned VT type sequences of AM fungi found in the desert samples and all AM fungal VT from Maarj*AM* database using neighbor-joining with *nj* function from the ape package in R. In order to represent phylogenetic community composition, we performed non-metric multidimensional scaling (NMDS) using a between-sample phylogenetic distance matrix. The distance matrix, representing mean phylogenetic distance (mpd) separating pairs of taxa in different samples, was based on AM fungal VT presence–absence and was constructed using the function *comdist* from the picante package [65]. Variation in AM fungal and plant community composition were visualized using principal coordinates analysis (PCoA) [66]. Sample dissimilarity was calculated using the Jaccard index based on the presence and absence of AM fungal VT or plant OTU. The environmental factors (annual mean temperature and precipitation, soil organic C, total N, mobile P and pH) were fitted to the ordination with the *envfit* function from the vegan package in R and visualized on the plot of NMDS or PCoA axis scores using arrows for AM fungal and plant data respectively. We investigated whether AM fungi present in desert samples represented random phylogenetic subsets of taxa from globally, continent or biogeographic realm. The global pool contained all the Maarj*AM* VT sequences while the desert available pool contained only VT recorded in this study. Mean pairwise phylogenetic distance (mpd) between VT for each sample was calculated using the function *ses.mpd* from R package picante.

We established whether the AM fungal taxa recorded in our samples had previously been recorded in desert and xeric shrublands using information from the Maarj*AM* database. For each VT we estimated an aridity index by calculating the fraction of its records in the Maarj*AM* database derived from “deserts and xeric shrublands”. A community-level aridity index was then estimated for each desert sample by calculating the mean aridity index of VT recorded in the sample. Random null models were then constructed for each sample by randomly selecting n VT, where n corresponds to the number of VT recorded in the empirical sample, among different pools of VT: All VT recorded in the Maarj*AM* database; or those VT previously recorded in the continent or biogeographic realm where the sample was collected. We repeated this procedure 999 times, each time calculating the mean aridity index for the randomly selected community. We compared the observed value and randomized values by calculating a Z-value (mean_observed-mean_randomized/standard deviation_randomized) to assess whether the representation of desert and xeric shrubland-affiliated VT in empirical samples was different from random expectation. Significant deviation from the random null model was indicated by *Z*-values more extreme than −1.96/1.96.

## 3. Results

The AM fungal dataset contained 542,129 quality filtered paired reads. 23,447 chimeric reads were found and removed from the analyses. BLAST resulted in 241,729 reads identified as AM fungi (containing 55 VT and 1 unpublished *Glomus* JD-GL07). Singletons were omitted from the BLAST results leaving a total of 49 VT and 1 unpublished (Appendix A). Therefore, 21 taxa (42%) were identified as cultured taxa. *Glomeraceae* was the most common family, while *Claroideoglomeraceae, Diversisporaceae* and *Acaulosporaceae* were represented with lower frequency and abundance. Sequences not getting a hit against Maarj*AM* database were subjected to identification with BLAST against GenBank. BLAST hits were distributed as following: 56% Metazoa (90% Collembola), 39% Fungi (77% Chytridiomycetes) and 1% Viridiplantae. No novel VT were identified among the non-AM fungal reads.

Analysis of AM fungal diversity showed that the Israeli site was the most diverse, and the Saudi Arabian site was the least diverse, whichever diversity measure was used (Table 3). Extrapolated Shannon and Simpson diversity estimates indicated that the United States site tended to be the second most diverse and the Kazakhstan site the third most diverse. Rarefaction curves showed that samples from Israel and Australia, perhaps also from Kazakhstan, had sufficient sequencing depth as they approached an asymptote, but others exhibited insufficient sequencing depth, suggesting that the extrapolated diversity estimates are likely to better capture the true diversity of the communities (Figure 2).

Many of the AM fungal taxa identified in this study have previously been recorded in arid biomes, but most are also widely distributed, both geographically and ecologically, occurring in several regions and biomes globally (Appendix A). The Saudi Arabian site was different from others—all three taxa recorded in this site were representatives of *Diversisporaceae* and all exhibit relatively narrow ecological distribution (Appendix A). A randomization test showed that “desert affiliated” VT (i.e., VT which have been, according to Maarj*AM* database, recorded from deserts and xeric shrublands in any part of the globe) were over-represented in the samples from Australia, Argentina, Israel and Kazakhstan, compared to records in the database originating either from the same continent or biogeographic realm (Table 4).

The phylogenetic composition of AM fungal communities (Figure 3) in samples formed three groups on the ordination biplot. The first group comprised samples from Australia, Argentina and Israel, which also exhibited the highest observed AM fungal taxon richness; the second group comprised samples from the USA and Kazakhstan, which both exhibited relatively high extrapolated fungal diversity and the third group consisted of a single sample from Saudi Arabia, which was extremely taxon poor. For comparison presence–absence based AM fungal VT ordination biplot (Appendix A) was also generated showing similar ordination as the AM fungal phylogenetic composition. AM fungi found in the three desert samples (AUS, ARG and KAZ) were phylogenetically more closely related than would be expected from random sampling of taxa from the global VT phylogeny (containing VT type sequences from the Maarj*AM* database, Appendix A).

## 4. Discussion

This study provides a first insight into AM fungal communities occurring in desert ecosystems across the globe. In general, the study site in Israel was the most taxon rich, while sites in Argentina, Australia, Kazakhstan and United States exhibited lower richness and diversity. Only three AM fungal taxa, all from the family *Diversisporaceae*, were recorded in a Saudi Arabian site. This site was also characterized by the highest mean annual temperature and the lowest annual precipitation rate. Most of the AM fungal taxa found in desert soils are predominantly widespread and occur in a wide range of biome types globally. Virtual taxa that have been previously recorded in deserts or xeric shrublands were over-represented in four of the six studied sites. VT identified from the samples were phylogenetically clustered compared with global taxon pool, suggesting that nonrandom assembly processes, notably habitat filtering, may have shaped fungal assemblages.

Our hypothesis that the diversity of AM fungi in desert ecosystems is low was only partially confirmed. The number of AM fungal taxa recorded was very low under the harshest conditions of high temperature and low precipitation. The total number of VT recorded in other desert sites was still below the average of around 50 VT per site among Maarj*AM* database records [67]. Our result is broadly consistent with previous spore-based studies showing variation in AM fungal diversity between different habitat types within desert landscapes and the recording of more taxa in benign sites [25,26,27]. At the Israeli site, AM fungal diversity was unexpectedly high despite the abiotic conditions being fairly similar to those at other sites. This may be related to the enhanced local dispersal of AM fungi due to grazing or some other past human impact, but further study would be needed to understand diversity patterns in the region. At the same time sequencing depth at some of the sites was insufficient to describe AM fungal richness, meaning that comparisons of richness with previous studies should be made with caution.

The phylogenetic and taxonomic composition of AM fungal communities was primarily driven by soil pH. This finding supports emerging evidence that pH plays a central role in the assembly of multiple soil communities [68]. Likely mechanisms underlying pH effects on microbial communities include mediation of nutrient availability. pH has a major impact of the mobility of multiple compounds, and hence on many connected biological processes in soil [69,70]. The impact of climatic factors, which have been considered the main global drivers of fungal community composition [71], appeared to cause the contrast between the Saudi Arabian site and other sites. The effect of soil factors on AM fungal communities is in line with previous work [10,71].

Although information about the mycorrhizal growth response of plant species is accumulating [72], there is no information about the responses of those species that were common in our study sites. Two sites—Argentina and Australia—were notable due to the relatively high proportion of grasses in the vegetation. The plant species also differed in terms of their photosynthetic pathways. In general, C4 grasses tend to respond more positively to mycorrhizal inoculation than C3 grasses [73]. However, Worschel et al. [18] found the opposite pattern under drought conditions, and suggested that the greatest benefit of AM fungi for C4 grasses may be increased nutrient acquisition, while for C3 grasses it may be water acquisition. Indeed, the two sites with the highest mean annual precipitation (MAP), Argentina and Australia, comprised more C4 species among common plants, while C3 species dominated in drier sites. However, whether this difference could be related to mycorrhiza in any way would need further investigation. The Saudi Arabian site was again exceptional, in that the only recorded grass was a C4 species. Previous work has recorded several unique AM fungal taxa in desert ecosystems [25], and we hypothesized that certain AM fungal taxa are characteristic of desert ecosystems and are rare in other biomes. This was not strongly supported by data. Desert AM fungal community composition varied according to abiotic conditions, but the fungal taxa recorded were mostly widespread. However, many of the recorded taxa had previously been recorded in dry habitats according to the Maarj*AM* database, and an abundant taxon in three of our study sites—VT388 (*Glomus* sp.)—was a dominant taxon in a xeric shrubland in Spain [74]. Moreover, based on an analysis of Maarj*AM* database records, the fungal taxa recorded at four of the six sites in this study exhibited a greater affiliation for desert and xeric shrubland ecosystems than might be expected from a random selection of VT. AM fungal VT identified from the desert samples also exhibited phylogenetic clustering, which may suggest that habitat filtering shaped the fungal assemblages [75]. In the Saudi Arabian site only three *Diversispora* taxa with narrow geographic and ecological distributions were recorded. Former studies have reported the presence of *Diversispora* taxa in the Arabian desert [20], but also in the early stages of primary succession [76,77] and in disturbed ecosystems in human impacted landscapes [78,79]. The Saudi Arabian site has a very sparse plant community, and might, to some extent, resemble an early successional ecosystem. It is also located in the central part of a large desert region, so certain fungal taxa may have not reached the study site due to dispersal limitation. *Diversispora* generally have small spores, which might also allow relatively efficient aerial dispersal [24]. Further studies are required to investigate the affiliation of *Diversispora* taxa with desert habitats.

AM fungal taxa exhibit differences in their ability to produce spores that facilitate long distance dispersal [80]. Only AM fungal taxa that produce spores can be multiplied with the current methods used to bring AM fungi to culture. Cultured AM fungal taxa are thus expected to be better colonizers than uncultured taxa [80]. Among the VT recorded in desert sites, 42% represented cultured taxa. This percentage is higher that common in natural habitats and correspond to what has been recorded in disturbed unwooded sites [81]. One of the reasons for this may be the better ability of cultured taxa to spread, which allows to inhabit areas with sparse vegetation and partly also unstable substrates.

The current study provides a first insight into desert soil AM fungal communities, based on environmental DNA metabarcoding. Our results show that desert AM fungal community composition and diversity vary according to the ecological conditions. Low diversity and in some cases also specific AM fungal community composition are likely to be related to harsh abiotic conditions and perhaps also to dispersal limitation in large desert areas. As climate change is likely to lead to desertification in many regions, more data on AM fungi as important mutualists should be collected to understand and predict ecosystem change. If AM fungal taxa well suited to desert conditions are identified, their use as inoculum could also be considered, for example in ecosystem restoration. Cultured taxa of AM fungi, which are common in deserts, may be the first candidates for use as an inoculum in the restoration of degraded arid ecosystems.

## Figures and Tables

**Figure 1 microorganisms-09-00229-f001:**
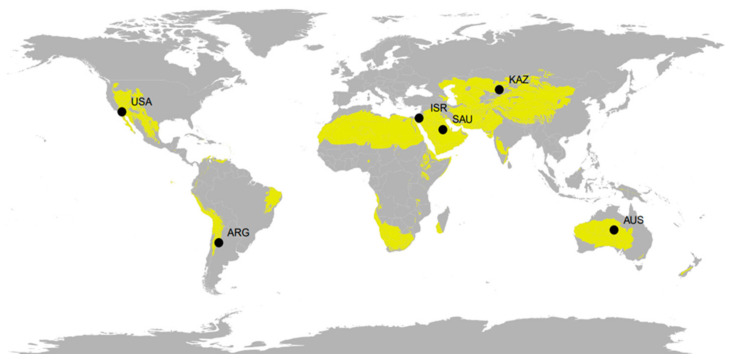
Sampling locations and desert and arid shrubland biomes colored yellow.

**Figure 2 microorganisms-09-00229-f002:**
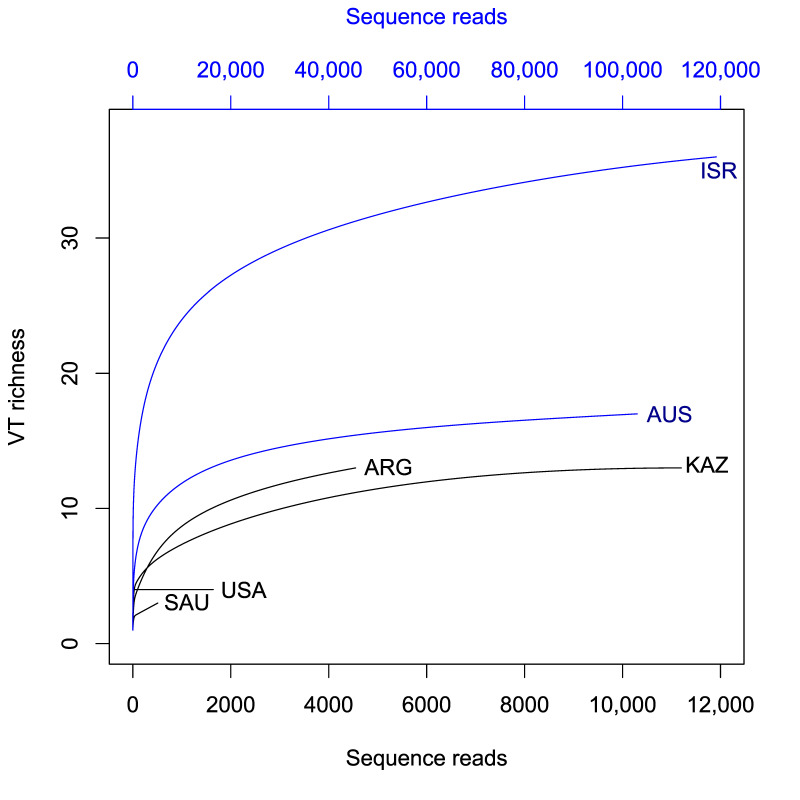
Rarefaction curves for study sites—accumulation of AM fungal VT in relation to the number of reads. Blue lines use scaling of axis on the top. Part of the samples AUS and ISR curves are cut-off to fit the graph.

**Figure 3 microorganisms-09-00229-f003:**
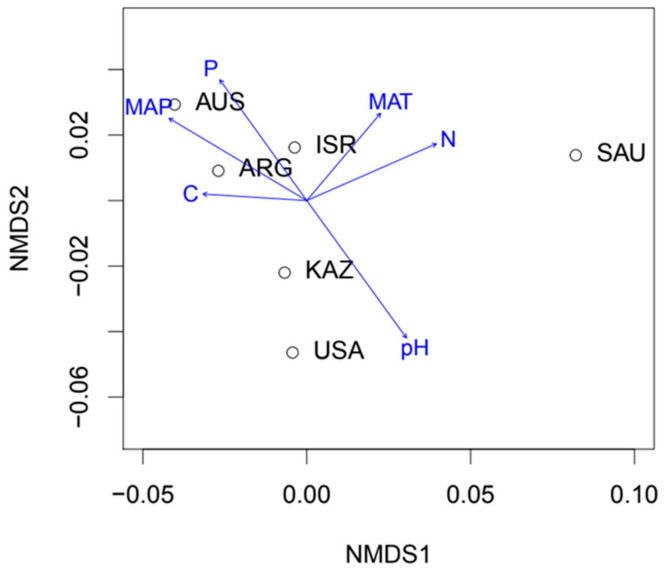
NMDS ordination showing variation in AM fungal phylogenetic community composition for desert samples. The ordination was calculated using mean pairwise phylogenetic distance between VT in samples. Arrows show significant correlations of climatic and soil–chemical parameters with the NMDS configuration.

**Table 1 microorganisms-09-00229-t001:** Study site characteristics. Mean annual temperature (MAT), mean annual precipitation (MAP), soil organic carbon (C), total nitrogen (N), available phosphorus (P), Argentina (ARG), Australia (AUS), Israel (ISR), Kazakhstan (KAZ), Saudi Arabia (SAU) and United States of America (USA). Ecoregion is from [36].

Site	Country, District and the Settlement Nearest to the Site	Ecoregion	Coordinates	MAT (°C)	MAP (mm)	pH	P (mg/100 g Soil)	N (%)	C (%)
ARG	Argentina, La Rioja, Los Colorados	Dry chaco	29.58,11 S 67.5,29 W	19.2	319	6.40	11.40	0.03	0.60
AUS	Australia, Northern territory, Alice Springs	Central Ranges xeric scrub	23.46,01 S 133.52,32 E	22.4	278	4.90	61.00	0.03	0.29
ISR	Israel, Southern district, Mitzpe Ramon	Mesopotamian shrub desert	30.36,35 N 34.44,31 E	17.1	113	7.47	0.38	0.09	0.62
KAZ	Kazakhstan, Zhambyl district, Taukum desert	Central Asian northern desert	44.24,41 N 75.31,15 E	10.6	192	7.59	1.98	0.03	0.42
SAU	Saudi Arabia, Riyadh, Arabian desert	Arabian desert	24.58,58 N 46.50,60 E	25.9	105	8.50	22.60	0.10	0.45
USA	United States of America, California, Boyd	Sonoran desert	33.39,02 N 116.22,30 W	21.2	141	7.35	6.51	0.05	0.37

**Table 2 microorganisms-09-00229-t002:** Life zone and common plant species in study sites. Taxonomy follows “The Plant List” (www.theplantlist.org). Abbreviation spp is used when it was possible to recognize only genus. Dominants are indicated in bold. Life zone is from [37]. The C3, C4, and CAM (Crassulacean acid metabolism) photosynthesis models are based on a common photosynthetic core with additional fluxes to capture the spatial and temporal separations of CO_2_ uptake and fixation.

Site	Life Zone	Vegetation Coverage (%)	Common Species	Growth Form	Photosynthetic Pathway
ARG	Warm temperate desert scrub	20	***Larrea cuneifolia***	Shrub	C3
			***Opuntia articulata***	Forb	CAM
			***Zuccagnia punctata***	Shrub	C3
			*Atriplex lampa*	Shrub	C4
			*Bouteloua aristidoides*	Grass	C4
			*Cottea pappophoroides*	Grass	C4
			*Cyclolepis genistoides*	Shrub	C3
			*Gymnocalycium schickendantzii*	Forb	C3
			*Neobouteloua lophostachya*	Grass	C4
			*Pappophorum philippianum*	Grass	C4
			*Porophyllum obscurum*	Forb	C3
			*Prosopis chilensis*	Tree	C3
AUS	Subtropical desert scrub	60–80	***Cenchrus ciliaris***	Grass	C3 and C4
			*Acacia spp*	Tree	C3
			*Aristida contorta*	Grass	C3 and C4
			*Triraphis mollis*	Grass	C3 and C4
			*Eragrostis barrelieri*	Grass	C4
			*Calocephalus platycephalus*	Forb	C3
			*Wahlenbergia spp*	Forb	C3
ISR	Warm temperate desert scrub	10	*Asphodelus ramosus*	Forb	C3
			*Erodium crassifolium*	Forb	C3
			*Helianthemum viscarium*	Forb	C3
			*Plantago afra*	Forb	C3
			*Ballota undulata*	Forb	C3
			*Pterocephalus brevis*	Forb	C3
KAZ	Cool temperate desert scrub	20	***Hordeum spontaneum***	Grass	C3
			***Bassia prostrata***	Shrub	C3
			***Heliotropium arguzioides***	Forb	C3
			*Artemisia semiarida*	Shrub	C3
			*Artemisia campestris*	Forb	C3
			*Eremurus inderiensis*	Forb	C3
			*Allium tulipifolium*	Forb	C3
			*Ceratocarpus arenarius*	Forb	C3
			*Astragalus maximowiczii*	Shrub	C3
			*Ammodendron bifolium*	Shrub	C3
			*Agropyron fragile*	Grass	C3
			*Bromus tectorum*	Grass	C3
			*Buglossoides arvensis*	Forb	C3
			*Consolida camptocarpa*	Forb	C3
			*Calligonum aphyllum*	Shrub	C4
SAU	Subtropical desert	<5	*Stipagrostis plumosa*	Grass	C4
			*Lasiurus scindicus*	Grass	C4
USA	Warm temperate desert scrub	30	*Larrea tridentata*	Shrub	C3
			*Ferocactus cylindraceus*	Stem succulent	C3
			*Opuntia littoralis*	Stem succulent	C3
			*Parkinsonia florida*	Tree	C3
			*Cylindropuntia bigelovii*	Stem succulent	C3
			*Salvia apiana*	Shrub	C3
			*Condea emoryi*	Shrub	C3

**Table 3 microorganisms-09-00229-t003:** AM fungal virtual taxa (VT) diversity at study sites—ARG, AUS, ISR, KAZ, SAU and USA. Observed richness and extrapolated values of richness, Shannon and Simpson diversity are given. Extrapolation was implemented in the iNEXT software [62]. Standard error (S.E.), lower confidence limits (LCL) and upper confidence limits (UCL).

		ARG	AUS	ISR	KAZ	SAU	USA
Species richness	Observed	13.0	17.0	36.0	13.0	3.0	4.0
	Estimated	17.5	18.0	37.6	13.0	3.0	4.0
	S.E.	7.19	2.3	2.16	0.74	0.48	0.0
	LCL	13.5	17.07	36.22	13.0	3.0	4.0
	UCL	53.91	31.25	47.79	15.16	4.5	4.0
Shannon diversity	Observed	1.92	1.89	6.6	2.61	1.34	3.8
	Estimated	1.92	1.89	6.6	2.61	1.34	3.81
	S.E.	0.03	0.01	0.02	0.02	0.05	0.03
	LCL	1.92	1.89	6.6	2.61	1.34	3.8
	UCL	1.98	1.91	6.64	2.66	1.43	3.87
Simpson diversity	Observed	1.46	1.51	4.52	1.9	1.18	3.62
	Estimated	1.46	1.51	4.52	1.9	1.18	3.63
	S.E.	0.02	0.01	0.02	0.02	0.03	0.06
	LCL	1.46	1.51	4.52	1.9	1.18	3.62
	UCL	1.5	1.52	4.56	1.94	1.23	3.74

**Table 4 microorganisms-09-00229-t004:** Results of randomization tests. Values in the table are *Z* values, where *Z* > 1.96 (bold) indicates significant positive deviation in the community aridity index compared with a random null model. The random null model consisted of sampling VT present globally, or present in the continent or biogeographic realm corresponding to the sample location.

**Scale**	**ARG**	**AUS**	**ISR**	**KAZ**	**SAU**	**USA**
All VT	1.560	1.337	1.612	1.551	−0.406	−0.331
Continent VT	**3.217**	**3.088**	**2.970**	**2.427**	−0.554	−0.166
Realm VT	**3.251**	**3.180**	1.254	1.417	−0.447	−0.152

## Data Availability

Data is contained within the article or Appendix A.

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
