# Peer review of "Arbuscular Mycorrhizal Fungal Communities in the Soils of Desert Habitats"

_microorganisms, 2021, doi:10.3390/microorganisms9020229_

Round 1

Reviewer 1 Report

The manuscript "Arbuscular Mycorrhizal Fungal Communities in the Soils of Desert Habitats", by Vasar and colleagues, is aimed at the characterisation of micorrhizal fungi colonising various desert areas. The Authors conclude that most of the fungal taxa have a broad distribution depending on the soil properties, while only few desert-specific taxa can be identified.

The work has a descriptive, rather than a probative accent. Its value may be greatly enhanced if the Authors' findings could be related to desert ecology, for instance by mentioning the ecological functions likely to be provided, or the plant species that may form symbiotic interactions with the described organisms.

The Authors do mention (but do not show) a study on plant community composition, which shows no correlation with the fungi; however, it may be worth trying if a change in perspective (from a population-based statistics to the targeted observation of relevant plant hosts) could improve the significance of the conclusions.

For the above reasons, this work has a potentially high interest to readers, but it should expanded in some of its aspects to be acceptable for publication.

A few minor issues and detailed comments are included in the attached file. The manuscript does not follow the Journal formatting style. Some supplementary information seems not to be included in the attached archive: figs S1, S2, S3; table S1.

Author Response

The manuscript "Arbuscular Mycorrhizal Fungal Communities in the Soils of Desert Habitats", by Vasar and colleagues, is aimed at the characterisation of micorrhizal fungi colonising various desert areas. The Authors conclude that most of the fungal taxa have a broad distribution depending on the soil properties, while only few desert-specific taxa can be identified.

The work has a descriptive, rather than a probative accent. Its value may be greatly enhanced if the Authors' findings could be related to desert ecology, for instance by mentioning the ecological functions likely to be provided, or the plant species that may form symbiotic interactions with the described organisms.

*We are grateful for this suggestion. We have added more information on this topic (cf. 374-381, 339-401).

AM fungal taxa exhibit differences in their ability to produce spores that facilitate long distance dispersal (Chagnon et al., 2013). Only AM fungal taxa that produce spores can be multiplied with the current methods used to bring AM fungi to culture. Cultured AM fungal taxa are thus expected to be better colonizers than uncultured taxa (Chagnon et al., 2013). Among the VT recorded in desert sites, 42% represented cultured taxa. This percentage is higher that common in natural habitats and correspond to what has been recorded in disturbed unwooded sites (Garcia de Leon et al. 2018). One of the reasons for this may be the better ability of cultured taxa to spread, which allows to inhabit areas with sparse vegetation and partly also unstable substrates.

Cultured taxa of AM fungi, which are common in deserts, may be the first candidates for use as an inoculum in the restoration of degraded arid ecosystems.

The Authors do mention (but do not show) a study on plant community composition, which shows no correlation with the fungi; however, it may be worth trying if a change in perspective (from a population-based statistics to the targeted observation of relevant plant hosts) could improve the significance of the conclusions.

*We have now included information about the plant community species composition at study sites (Table 2). We also removed the statement about correlation between plant and fungal communities that was not supported by robust empirical data (line 378-390). We attempted to characterize the plant community with the help environmental DNA by sequencing the trnL amplicon as suggested by Yoccoz et al. (2012), but the quality of the sequencing data was unfortunately insufficient, and so we do not presented them in the Ms.

A few minor issues and detailed comments are included in the attached file.

*We are very grateful for these suggestions and have corrected the Ms accordingly, cf. line 17, 36, 51, table 1, line 142, line 166-169, 187, 195, 275, 316-318, 385-390. Response to the reviewed pdf also attached.

The manuscript does not follow the Journal formatting style.

*We have now formatted the references.

Some supplementary information seems not to be included in the attached archive: figs S1, S2, S3; table S1.

*We have now uploaded the supplementary information.

Reviewer 2 Report

Here is the review of the manuscript entitled "Arbuscular Mycorrhizal Fungal Communities in the Soils of Desert Habitats".

The object of the paper was to research the diversity of arbuscular mycorrhizal (AM) fungi in six deserts (Argentina, Australia, Israel, Kazakhstan, Saudi Arabia, USA). Data on AM fungal diversity were obtained by high-throughput sequencing (Illumina MiSeq) of soil samples using the primer pair WANDA-AML2. Soil parameters szuch as pH, total N and organic C, as well as content of plant available P, K, Mg, and Ca in soils were measured and analysed. Total of 50 AM fungal phylotypes (virtual taxa, VT) excluding singletons were recorded in the study. Glomeraceae was the most common family, while Claroideoglomeraceae, Diversisporaceae and Acaulosporaceae were represented with lower frequency and abundance. Israeli Negev desert was the most taxon rich site, with 36 VT. Saudi Arabian site was the least diverse with only 3 VT recorded. AM fungal VT found were phylogenetically clustered and compared with the global taxon pool. The conclusion was that habitat filtering may have shaped desert fungal assemblages studied.

The paper is scientifically sound. The research methods are suitable and well conducted. There are some minor corrections that are proposed. I am recommending the paper for publication after the minor correction.

Please see the list of my remarks:

85 30m -> 30 m

112 topsoil (coment: How deep?)

137 Retch -> Retsch

183 vsearch -> Vsearch

245 and -> and

252 taxa -> taxon

277 been recorded -> recorded

Sampling sites are not adequately described, please state at least names and geographic coordinates of sampling sites. If needed refer to:

Leal & al. (2016) Natural products discovery needs improved taxonomic and geographic information. Nat. Prod. Rep. 33: 747-750.

Bloom & al. (2017) Why georeferencing matters - Introducing a practical protocol. Ecology and Evolution, 8: 765-777.

Figure 4. (Please discuss environmental parameters: MAT – mean annual temperature, MAP – mean annual precipitation, C - soil organic carbon; N - total nitrogen; P - available phosphorus). Also you state (line 355) that pH is one of the important drivers (but you did not explain in which way?).

Results & Table S2: 50 VT discovered.

Figure S1: 51 VT.

Check the numbers (should be equal).

References are not numbered in the text.

Best,

Reviewer

Author Response

The object of the paper was to research the diversity of arbuscular mycorrhizal (AM) fungi in six deserts (Argentina, Australia, Israel, Kazakhstan, Saudi Arabia, USA). Data on AM fungal diversity were obtained by high-throughput sequencing (Illumina MiSeq) of soil samples using the primer pair WANDA-AML2. Soil parameters szuch as pH, total N and organic C, as well as content of plant available P, K, Mg, and Ca in soils were measured and analysed. Total of 50 AM fungal phylotypes (virtual taxa, VT) excluding singletons were recorded in the study. Glomeraceae was the most common family, while Claroideoglomeraceae, Diversisporaceae and Acaulosporaceae were represented with lower frequency and abundance. Israeli Negev desert was the most taxon rich site, with 36 VT. Saudi Arabian site was the least diverse with only 3 VT recorded. AM fungal VT found were phylogenetically clustered and compared with the global taxon pool. The conclusion was that habitat filtering may have shaped desert fungal assemblages studied.

The paper is scientifically sound. The research methods are suitable and well conducted. There are some minor corrections that are proposed. I am recommending the paper for publication after the minor correction.

*Thank you very much for generous words!

Please see the list of my remarks:

85 30m -> 30 m  

*A change made as suggested, cf. line 81

112 topsoil (comment: How deep?)

*A change made as suggested, cf. line 107 (0-5cm)

137 Retch -> Retsch

*A change made as suggested, cf. line 136

183 vsearch -> Vsearch

*A change made as suggested, cf. line 184

245 and -> and

*A change made as suggested, cf. line245

252 taxa -> taxon

*This figure is omitted in a new version of the Ms

277 been recorded -> recorded

*A change made as suggested, cf. line 275

Sampling sites are not adequately described, please state at least names and geographic coordinates of sampling sites.

*We acknowledge the criticism. We have now added information about the location of the sampling sites, together with geographic coordinates, to Table 1.

Figure 4. (Please discuss environmental parameters: MAT – mean annual temperature, MAP – mean annual precipitation, C - soil organic carbon; N - total nitrogen; P - available phosphorus). Also you state (line 355) that pH is one of the important drivers (but you did not explain in which way?).

*Many thanks for important suggestion. We have added discussion and explanation (lines 330-338).

The phylogenetic and taxonomic composition of AM fungal communities was primarily driven by soil pH. This finding supports emerging evidence that pH plays a central role in the assembly of multiple soil communities (Glassman et al. 2017). Likely mechanisms underlying pH effects on microbial communities include mediation of nutrient availability. PH has a major impact of the mobility of multiple compounds, and hence on many connected biological processes in soil (Tyler & Olsson 2001, Neina 2019). The impact of climatic factors, which have been considered the main global drivers of fungal community composition (Tedersoo et al. 2014), appeared to cause the contrast between the Saudi Arabian site and other sites. The effect of soil factors on AM fungal communities is in line with previous work (Davison et al. 2015, Tedersoo et al. 2014).

Results & Table S2: 50 VT discovered.

Figure S1: 51 VT.

Check the numbers (should be equal).

* Thank you for pointing out the problem. The right number is 49 VT + 1 unpublished, so 50 phylotypes in total. Table S2 included all the 50 phylotypes. Figure S1 and S2 presented biome occurrence of the 49 VT that have previously been recorded. Please see the revised supplementary doc (Figure S1).

References are not numbered in the text.

 *We have now formatted the references.

Round 2

Reviewer 1 Report

The manuscript has been modified according to this Reviewer's suggestion and expectation, and deserves publication in the present form. Please find a few minor edits in attachment.
